# Protective Effects of Velvet Antler Methanol Extracts on Hypoxia-Induced Damage in *Caenorhabditis elegans* through HIF-1 and ECH-8 Mediated Lipid Accumulation

**DOI:** 10.3390/nu16142257

**Published:** 2024-07-13

**Authors:** Ru Li, Hongyuan Li, Xiaohui Wang, Yinghua Peng

**Affiliations:** 1Institute of Special Animal and Plant Sciences, Chinese Academy of Agricultural Sciences, Changchun 130112, China; ru.li@ciac.ac.cn; 2Laboratory of Chemical Biology, Changchun Institute of Applied Chemistry, Chinese Academy of Sciences, Changchun 130022, China; hongyuan.li@ciac.ac.cn; 3School of Applied Chemistry and Engineering, University of Science and Technology of China, Hefei 230026, China

**Keywords:** velvet antler, hypoxia, HIF-1, lipid accumulation, *C. elegans*

## Abstract

Velvet antler, a traditional tonic widely used in East Asia for its health benefits, is explored in this study for its protective effects against hypoxia-induced damage using *Caenorhabditis elegans* (*C. elegans*) as a model. Hypoxia, characterized by low oxygen availability, induces significant physiological stress and potential tissue damage. Our research demonstrates that methanol extracts from velvet antler (MEs) enhance the survival of *C. elegans* under hypoxic conditions. This enhancement is achieved through the stabilization of hypoxia-inducible factor-1 (HIF-1) and the promotion of lipid accumulation, both of which are crucial for mitigating cellular damage. Specifically, MEs improve mitochondrial function, increase ATP production, and aid in the recovery of physical activity in *C. elegans* post-hypoxia or following hypoxia–reoxygenation (HR). The pivotal role of HIF-1 is underscored by the loss of these protective effects when HIF-1 function is inhibited. Additionally, our findings reveal that the gene related to lipid metabolism, *ech-8*, significantly contributes to the lipid accumulation that enhances resilience to hypoxia in *C. elegans* treated with MEs. These results not only highlight the therapeutic potential of velvet antler in modern medical applications, particularly for conditions involving hypoxic stress, but also provide insights into the molecular mechanisms by which MEs confer protection against hypoxic damage.

## 1. Introduction

Oxygen is essential for the survival and efficient functioning of aerobic organisms, which rely on it for ATP synthesis [1]. To balance oxygen availability with metabolic demand, these organisms have developed complex adaptive systems [2]. A significant challenge in maintaining this balance is hypoxia, a condition where oxygen levels are too low for normal tissue function, leading to physiological stress, tissue injury, and impaired cellular functions [3]. Evolutionary adaptations have equipped organisms with mechanisms to mitigate the effects of hypoxia, notably through the activation of the hypoxia-inducible factor-1 (HIF-1) pathway [4]. HIF-1 serves as a transcriptional regulator, controlling genes that facilitate adaptation to low oxygen environments. These adaptations include changes in metabolic processes [5], angiogenesis [6], and erythropoiesis [7], which are crucial for survival under hypoxic conditions. Disruptions in oxygen homeostasis are associated with various age-related diseases, including cancers and neurodegenerative disorders [8,9], underscoring the importance of modulating these hypoxic defenses for developing targeted therapies.

In recent years, traditional medicines, known for their health benefits and minimal side effects, have garnered attention. Velvet antler, a revered traditional Chinese medicinal ingredient, comprises a distinctive type of cartilaginous tissue enveloping the developing antlers of several deer species, such as red deer, elk, and moose. This dietary supplement holds esteemed status in Asian cultures due to its purported health benefits and anti-aging properties, attracting considerable scientific interest and serving as a focal point for numerous research efforts [10]. Research has shown that methanol extracts from velvet antler (MEs), which primarily contain terpenoids, phenols, steroids, lipids, and glycosides, can counteract oxidative stress and alleviate symptoms of Parkinson’s disease (PD) in model organisms like *C. elegans* and mice [11,12]. This study aims to evaluate the effectiveness of MEs in reducing hypoxic injury in *C. elegans* and to elucidate the protective mechanisms against hypoxia-induced damage.

The molecular mechanisms regulating the hypoxia response in *C. elegans* are highly conserved across mammals, with HIF-1 playing a crucial role [13]. Under normal oxygen levels (21% O_2_), HIF-1 is inactive and rapidly degraded [14]. However, under hypoxic conditions, HIF-1 is stabilized, triggering a gene expression program tailored for hypoxic stress adaptation [15]. Our findings reveal that MEs enhance the survival of *C. elegans* by modulating HIF-1 and altering fat metabolism to increase lipid storage, thereby reducing damage from hypoxia. The gene *ech-8*, an ortholog of human EHHADH (enoyl-CoA hydratase and 3-hydroxyacyl CoA dehydrogenase), plays a crucial role in fatty acid metabolism, particularly in the catabolism of fatty acids to meet energy demands during fasting or increased metabolic activity [16]. It is located on chromosome IV (10236626–10239176) and comprises eight exons. The expression of this gene varies across different developmental stages and in response to nutritional status. Research has indicated that knock down of the *ech-8* gene could result in metabolic disorders affecting fat content [17].

Additionally, the gene *ech-8* has been identified as instrumental in lipid accumulation, further aiding in the protection provided by MEs under hypoxic conditions. These results highlight the potential of MEs as an effective treatment for hypoxic injuries and emphasize the importance of exploring traditional medicines for modern therapeutic applications.

## 2. Materials and Methods

### 2.1. Strains and Antler Velvet Extracts

All strains of *C. elegans* were cultivated at a temperature of 20 °C and fed *Escherichia coli* (*E. coli*) OP50 on nematode growth media (NGM) unless otherwise indicated. The NGM were sterilized in an autoclave and then carried out in a sterile super-clean table to ensure that there was no infection. The Bristol strain N2 was used as the wild-type strain; strains SJ4103 (zcIs14 [myo-3::GFP(mit)]) and AR7 (hif-1(mr22)V) were obtained from the Caenorhabditis Genetic Center (CGC) and used in this study.

Velvet antler from sika deer (*Cervus Nippon*) was provided by the Zuojia Sika Deer Farm (Jilin, China). Velvet antler methanol extracts (MEs) were prepared as previously described [12]. MEs were added to the *C. elegans* culture at the L1 larval stage. Specifically, the extract was incorporated into the nematode growth medium (NGM) plates at a final concentration of 0.1, 0.4, and 1 mg/mL. The *C. elegans* were exposed to the ME from the L1 stage until they reached young adulthood to ensure continuous exposure and to assess the effects throughout their development.

### 2.2. Hypoxic and Hypoxia–Reoxygenation (HR) Incubation

Young adult worms, initially synchronized at the L1 stage, were subjected to hypoxic conditions by immersion in 500 µL of M9 buffer within a 48-well culture plate and exposed to a low-oxygen environment (0.1% O_2_, balanced with nitrogen) for 36 h at 26 °C in a specialized low-oxygen chamber (GC-C, Maworde Ltd. Qiqihar, Heilongjiang, China). Following this, a 24 h recovery period under standard atmospheric conditions on OP50 plates at 20 °C was provided to simulate HR incubation. The oxygen levels within the hypoxic chamber were consistently maintained and continuously monitored automatically.

### 2.3. Survival and Lifespan Analysis

For the survival assay, the worms were exposed to a hypoxic environment with 0.1% O_2_ for 36 h at 26 °C. After this hypoxia exposure, some groups also underwent reoxygenation treatment. To assess survival, the worms’ touch response was evaluated, with a positive response indicating survival and a lack of response indicating death. The daily survival rate was determined by gently stimulating the pharynx to observe a response. These behavioral assessments were repeated three times.

### 2.4. Locomotion and Feeding Behavior Analysis

Worms were rinsed with M9 buffer and subsequently transferred onto a glass slide with 100 μL of M9 buffer. Following a recovery period of 1 min, the number of body thrashes was recorded for 1 min. A thrash was defined as a movement in which the worm swings its head and/or tail to the same side, and each occurrence was counted as one thrash. The evaluation of feeding behavior involved measuring the pharyngeal pumping rate, which was determined by counting the pharyngeal contractions over a 30 s period. These behavioral tests were repeated three times with 15 randomly selected worms per treatment.

### 2.5. Mitochondrial Morphology

For the assessment of changes in mitochondrial morphology, SJ4103 worms with mitochondria-tagged GFPs in their body wall muscle (zcIs14 [myo-3::GFP (mit)]) were used. At the end of exposure hypoxia and HR, the worms were collected and anesthetized in 2 mM levamisole and mounted onto a microscope slide containing a 2% agarose pad. To prevent the artifacts, images were captured immediately after the preparation of slides using a Nikon TS2-FL fluorescence microscope with a 100× oil immersion lens. Mitochondrial morphology was blindly evaluated three times, with a minimum of 15 animals for each condition.

### 2.6. ATP Measurement

After hypoxia or HR treatment, the worms were washed with M9 buffer four times and then subjected to ultrasonication using an Ultrasonics Processor SY-150 (Shanghai Ninson Co., Ltd., Shanghai, China) to crush the worms. The resulting supernatant was collected following centrifugation (12,000 r/min, 5 min, 4 °C), and the ATP concentrations were determined at 560 nm using a luciferase-based enhanced ATP assay kit (Beyotime Biotechnology, S0027, Shanghai, China) as per the manufacturer’s instructions. Additionally, the protein concentration of the samples was assessed at 562 nm using a bicinchoninic acid (BCA) assay kit (Tiangen, PA115-01, Beijing, China). The measured ATP levels were then normalized to the protein concentration. This process was repeated independently three times with three replicates at each time point.

### 2.7. RNA Isolation and Quantitative PCR (qPCR)

After hypoxia or HR treatment, worms were washed with M9 buffer four times. Total RNA was extracted from the worms using a TRIzol reagent kit (Yeasen, 10606ES60, Shanghai, China) following the manufacturer’s instructions. To ensure RNA quality, we evaluated RNA quality and integrity by comparing the brightness and clarity of the three strips (28S, 18S, 5S) by agarose gel electrophoresis. In addition, the absorbance of RNA at 260 nm and 280 nm was measured using a microplate to obtain the ratio 260/280 and quantify the concentration of RNA. Subsequently, cDNA was synthesized using the Hifair III 1st Strand cDNA Synthesis SuperMix (Yeasen, 11141ES60, Shanghai, China) for qPCR according to the provided protocol. qPCR was performed using SYBR Green RT-PCR Master Mix (Yeasen, 11201ES03, Shanghai, China) on a TOptical RT-PCR system (Analytik Jena AG, Analytik Jena, Jena, Germany). The results were analyzed and presented as fold-change calculations (2^−ΔΔCt^) manually. The genes *cdc-42* and *y45f10d.4* were utilized as internal controls for gene expression normalization. The primer sequences for PCR are available in Table 1.

### 2.8. Oil Red O-Based Lipid Staining

Worms were collected following hypoxia. For the lipid staining experiments, a 0.5% stock solution of Oil Red O (BBI Life Science, A600395-0050, Shanghai, China) was prepared in high-quality isopropanol. The stock solution was then diluted to 60% with sterile water, incubated at room temperature overnight on a rocking platform, and filtered through a 0.45 µm filter for staining the worms. The worms were resuspended in 500 µL of 60% isopropanol for fixation. After removing the isopropanol, 500 µL of freshly filtered Oil Red O working solution was added, and the worms were incubated in a thermomixer at room temperature with gentle agitation for 10 min. Subsequently, the worms were washed three times with 500 µL of 0.01% Triton X-100 (Sigma-Aldrich, 9002-93-1, Darmstadt, Germany) in M9 buffer and then imaged on a Nikon TS2-FL fluorescence microscope using a 10× objective. The Oil Red O staining images were quantified and analyzed using Image J (Fiji 2.15.1).

### 2.9. RNA Interference (RNAi) Assay

*C. elegans* were raised on the *E. coli* strain HT115 harboring plasmids expressing dsRNA targeting specific genes. In the single-gene RNAi experiment, *E. coli* HT115 with an empty L4440 plasmid was used as a negative control in either *hif-1* or *ech-8* RNAi experiments. For the combined RNAi assay targeting both hif-1 and ech-8, *E. coli* HT115 (DE3) bacteria containing RNAi clones for *hif-1* and *ech-8* were mixed in a 1:1 ratio based on the absorbance of the bacterial cultures. Additionally, for the *hif-1* or *ech-8* RNAi alone groups, *E. coli* HT115 (DE3) bacteria containing RNAi clones for *hif-1* or *ech-8* were combined with *E. coli* HT115 containing an empty L4440 plasmid in a 1:1 ratio based on the absorbance of the bacterial cultures. This combination served as the control for the *hif-1* and *ech-8* combined RNAi group. Furthermore, *E. coli* HT115 with an empty L4440 plasmid alone was used as a negative control. These controls were generated using standard cloning techniques and were administered to *C. elegans* using the same feeding method as the experimental RNAi clones. To perform the RNAi assays for either *hif-1* or *ech-8*, the specified *E. coli* HT115 (DE3) bacteria containing RNAi clone for *hif-1* or *ech-8* were seeded onto NGM agar plates that had been supplemented with 50 μg/mL ampicillin and 1 mM *β*-D-1-thiogalactopyranoside (IPTG) and then stored for 24 h or longer for the bacterial lawns to grow. For the combined RNAi assay targeting both *hif-1* and *ech-8*, the specified *E. coli* HT115 (DE3) bacteria containing RNAi clones for *hif-1* or *ech-8* were combined in a 1:1 ratio based on the absorbance of the bacterial cultures. The mix was seeded onto NGM agar plates that had been supplemented with 50 μg/mL ampicillin and 1 mM IPTG and then stored for 24 h or more for the bacterial lawns to grow. Larvae at the L1 stage were then transferred onto fresh plates prepared for either *hif-1* or *ech-8* RNAi, or a combination of both. The efficacy of the RNAi was later assessed using qPCR.

### 2.10. Statistical Analysis

Statistical analysis was performed using GraphPad Prism 8 software (GraphPad Software, La Jolla, CA, USA). Data were presented as the mean ± SEM. Prior to performing statistical tests, normality of the data was confirmed using the Shapiro–Wilk test, provided there was a minimum of 3 independent replicates. For survival assay, locomotion assay, pharyngeal pumping assay, ATP levels assay, qPCR assays, and Oil Red O-based lipid staining assay, the Student’s *t*-test was utilized to analyze the difference between the two groups, and the one-way ANOVA method was applied to analyze the differences among groups. For lifespan assay, comparison was calculated using the log-rank (Mantel–Cox) test compared with the control group. For mitochondrial morphology, *p* values were determined by the chi-square test. The above experiment was repeated no less than three times and *p* < 0.05 was considered a significant difference.

## 3. Results

### 3.1. MEs Attenuate Damage Induced by Hypoxia in C. elegans

To investigate the potential protective effects of MEs on hypoxia-induced damage in *C. elegans*, N2 wild-type worms were treated with MEs for 60 h starting from the L1 larval stage. The efficacy of MEs in mitigating the adverse effects of hypoxia was evaluated by assessing the condition and mobility of the worms post-treatment. Initially, worms were exposed to a hypoxic environment containing 0.1% O_2_ at 26 °C, and their responses were documented (Figure 1A).

Following exposure to hypoxia, the majority of the control group worms remained immobile, although some began to move again after undergoing hypoxia–reoxygenation (HR) treatment. In the group treated with a low dose of MEs (0.1 mg/mL), the condition post-hypoxia was similar to that of the control group (Figure 1B,C). However, after 24 h of HR, a higher proportion of worms in the 0.1 mg/mL MEs group regained mobility compared to the control group. At higher dosages of MEs (0.4 and 1 mg/mL), there was a significant decrease in the number of immobile worms, with an increase in active worms both after hypoxia and following HR (Figure 1B,C).

These observations suggest that MEs enhance the survival and recovery of *C. elegans* under hypoxic conditions. To further assess the extent of hypoxia-induced damage and the protective effects of MEs, physiological functions such as the swallowing reflex and body movement were measured post-HR. Results showed a notable improvement in the swallowing frequency of worms treated with MEs compared to the HR control group (Figure 1D). Additionally, worms administered MEs demonstrated an increase in body bends, indicating enhanced motor activity post-HR (Figure 1E).

Long-term effects of MEs on survival were also evaluated by tracking the lifespan of *C. elegans* post-hypoxia (Figure 1F). While the administration of 0.1 mg/mL MEs did not significantly extend the lifespan compared to the control group, higher concentrations of MEs (0.4 and 1 mg/mL) markedly increased the long-term survival of the worms.

In conclusion, these results affirm that MEs play a crucial role in protecting *C. elegans* from hypoxic injury by enhancing both immediate and long-term survival and recovery. The results highlight the potential of MEs as a therapeutic intervention for conditions associated with hypoxic stress.

### 3.2. MEs Improve the Function of Mitochondria in Hypoxia-Treated C. elegans

To elucidate the mechanisms by which MEs enhance the anti-hypoxia effects in *C. elegans*, the function of mitochondria in hypoxia-treated worms was studied using the SJ4103 strain, which expresses high levels of GFP in mitochondria [18]. Given that mitochondria are the primary consumers of oxygen within cells and are thus highly susceptible to reduced oxygen availability, understanding their behavior under stress is crucial [19,20].

After exposure to hypoxia, the tubular mitochondria in *C. elegans* treated with MEs showed significant recovery compared to those in the vehicle-treated group, as observed in the mitochondrial morphology (Figure 2A,B). This recovery was also evident after 24 h of HR, where the tubular mitochondria fraction in the MEs-treated group showed notable improvement relative to the vehicle group (Figure 2A,B).

Further analysis using myo-3::GFP fluorescence expression as a marker for mitochondrial quantity revealed that MEs might increase the number of mitochondria in the worms. Post-hypoxia, the fluorescence expression in the MEs-treated worms was enhanced compared to the vehicle group (Figure 2C,D). However, this difference in fluorescence expression diminished after 24 h of HR, suggesting a normalization in mitochondrial count or function.

To validate the protective effects of MEs on mitochondrial integrity and function, ATP production was assessed. The ATP levels in worms treated with MEs showed a substantial increase both immediately after the hypoxic treatment (Figure 2E) and after 24 h of HR (Figure 2F). These findings suggest that MEs not only protect mitochondrial structure and function during hypoxic stress but also enhance ATP production in response to reduced oxygen availability.

In summary, these results demonstrate that MEs have a protective effect on mitochondria in *C. elegans*, improving their resilience and function under hypoxic conditions and potentially enhancing the overall anti-hypoxia effects.

### 3.3. MEs Protect against Hypoxia Damage by HIF-1 in C. elegans

Under hypoxic conditions, HIF-1 acts as a master regulator, adapting to low oxygen levels by inducing the expression of genes that enhance hypoxia resistance [21,22]. To investigate the role of HIF-1 in mediating the protective effects of MEs, experiments were conducted using *hif-1* mutants (AR7) and compared with N2 wild-type worms. The survival rate of *hif-1* mutants was significantly lower than that of N2 worms following hypoxia or HR (Figure 3A), consistent with previous findings that underscore HIF-1 as a crucial factor in hypoxia resistance in *C. elegans* [23]. Interestingly, MEs treatment did not improve survival in *hif-1* mutants compared to N2 worms under the same conditions, suggesting that HIF-1 is necessary for the protective benefits of MEs.

Further analysis focused on the mitochondrial dynamics in SJ4103 worms, which express high levels of GFP in mitochondria, treated with RNAi targeting *hif-1* (Appendix A). Post-hypoxia and HR, MEs treatment significantly enhanced the recovery of tubular mitochondria in the mock RNAi group compared to the vehicle treatment (Figure 3B,C). However, in the *hif-1* RNAi group, the MEs failed to restore the tubular mitochondrial fraction after hypoxia, though an increase was observed after HR (Figure 3C). This suggests that while HIF-1 influences mitochondrial morphology, its presence is essential for the full mitochondrial protective action of MEs.

In terms of energy production, MEs treatment increased ATP levels during hypoxia and post-HR in the mock RNAi group. This effect was absent in the *hif-1* RNAi knockdown group, where MEs did not influence ATP levels, reinforcing the critical role of HIF-1 in the energy-regulating effects of MEs during and after hypoxic stress (Figure 3D,E).

These findings collectively imply that HIF-1 is not only crucial for the morphological protection of mitochondria but also for the energy metabolism benefits conferred by MEs under hypoxic conditions. This underscores the importance of HIF-1 in the mechanism through which MEs mitigate hypoxia-induced damage in *C. elegans*.

### 3.4. MEs Protect against Hypoxia-Induced Damage via Promoting Lipid Storage

Hypoxia triggers a variety of adaptive cellular responses, one of which is a significant alteration in lipid metabolism [24]. During low-oxygen conditions, cells accumulate lipids as a protective mechanism [25]. This not only provides an energy reserve but also supplies essential components for membrane synthesis, helping to maintain crucial cellular functions and integrity, particularly when ATP production from oxygen-dependent pathways is reduced. As demonstrated in (Figure 4A,B), MEs enhance lipid accumulation in N2 worms more than in vehicle-treated N2 worms within the mock RNAi group. This suggests that MEs play a role in promoting lipid storage under hypoxic conditions.

ECH-8, or enoyl-CoA hydratase-8, is involved in the beta-oxidation pathway, critical for the breakdown of fatty acids [26,27,28]. The regulation of enzymes like ECH-8 becomes crucial in hypoxia, where rapid metabolic adjustments are necessary. When ECH-8 was knocked down in worms (Appendix A), these worms showed increased lipid accumulation compared to those treated with mock RNAi (Figure 4A,B). Notably, MEs did not enhance lipid accumulation in *ech-8* RNAi-treated worms compared to their vehicle-treated counterparts, suggesting that the effect of MEs might involve inhibition of ECH-8.

In contrast, knocking down HIF-1 did not increase lipid accumulation in N2 worms compared to the mock RNAi treatment (Figure 4C,D). Furthermore, lipid accumulation induced by MEs was reversed in HIF-1 RNAi-treated worms, indicating that HIF-1 function is necessary for MEs to promote lipid accumulation.

To further explore the interaction between ECH-8 and HIF-1 in regulating lipid accumulation, a combined RNAi assay targeting both genes was performed on N2 worms. Results shown in Figure 4E,F reveal that lipid accumulation was lower in the combined RNAi group than in the *ech-8* only RNAi group, suggesting that HIF-1 may moderate the impact of ECH-8 on lipid regulation. This finding is supported by the observed upregulation of *ech-8* expression in *hif-1* mutants (Appendix A).

The impact of HIF-1 and ECH-8 on the survival of N2 worms following hypoxia exposure was also measured. Results in Figure 4G show that *hif-1* RNAi decreased survival rates compared to mock RNAi, while *ech-8* RNAi increased survival rates. However, the survival rate was lower in the combined *hif-1* and *ech-8* RNAi-treated worms compared to those treated only with *ech-8* RNAi. This suggests that the protective effect against hypoxia via *ech-8* loss of function is dependent on the presence of HIF-1.

## 4. Discussion

Velvet antler has long been recognized in East Asian medicinal practices for its diverse health benefits, including anti-aging properties and enhancements in physical strength and vitality [11,29,30]. This study delves into the pharmacological actions of MEs in a hypoxia model, revealing their ability to mitigate damage and enhance survival by modulating specific genetic pathways associated with lipid metabolism. The findings provide compelling evidence of the significant protective effects of MEs against hypoxia-induced damage in *C. elegans*. Key mechanisms through which MEs exert these effects include the modulation of HIF-1 activity and the regulation of lipid metabolism, both of which are essential for the survival and recovery of organisms under hypoxic conditions.

The central role of HIF-1 in mediating the beneficial effects of MEs is critical, aligning with prior research on its regulatory capacity across various species and models in response to hypoxia [31,32]. In prior research, HIF-1 has been identified as a lipid-binding protein whose functionality critically relies on lipid reserves under hypoxic conditions [33]. Furthermore, HIF-1 modulates hypoxia-induced lipolysis in *C. elegans* by inhibiting PKA and ATGL proteins, emphasizing its central role in nematode lipid metabolism during oxygen deprivation [34]. In our study, RNAi-mediated suppression of HIF-1 during hypoxia markedly decreased lipid content, accompanied by a corresponding reduction in in vivo ATP metabolism.

This study demonstrates that under hypoxic conditions, the stabilization of HIF-1 activates a gene expression program essential for adaptation and survival. It suggests that MEs enhance the stability of HIF-1, leading to improved mitochondrial function and increased ATP production post-hypoxia. Significantly, the absence of HIF-1 in mutant strains eliminates the protective effects of MEs, highlighting its pivotal role in their mechanism of action.

Lipid accumulation appears to be a critical adaptive response to hypoxia facilitated by MEs. This process appears to be regulated through the sophisticated modulation of lipid storage, particularly via the action of the enzyme ECH-8. Lipid metabolism significantly influences the body’s nutritional balance during processes such as fasting and hypoxia. In *C. elegans*, lipid storage metabolism critically impacts survival [35,36]. Disruption of the lipid oxidation pathway was observed to affect lipid droplet formation, which is crucial for lipid storage and membrane fluidity.

Notably, inhibiting ECH-8 function results in enhanced lipid accumulation, indicating that lipid storage mechanisms are actively managed under stress conditions. Furthermore, it was observed that MEs did not increase lipid accumulation in N2 worms treated with ECH-8 RNAi. These findings support the hypothesis that during hypoxia, lipids serve a dual role by acting as energy reserves and as essential components for maintaining cellular integrity, thereby highlighting the complex interaction between MEs and lipid metabolism in hypoxic conditions.

The ability of MEs to modulate both mitochondrial function and lipid metabolism presents a novel therapeutic avenue for treating hypoxic injuries, which are common in numerous clinical conditions such as stroke, heart attack, and chronic lung diseases. The findings from this study suggest that traditional medicines like velvet antler contain bioactive compounds that could be harnessed to develop new treatments for diseases where hypoxia is a factor. Additionally, the favorable safety profile and minimal side effects associated with MEs underscore their potential for translation into clinical applications.

Despite the promising results, further research is necessary to fully understand the complex interactions and molecular mechanisms underlying the protective effects of MEs. Future studies should focus on isolating specific bioactive components within velvet antler that are responsible for the observed effects. Extending these investigations to mammalian models is crucial for validating the efficacy and safety of these compounds in higher organisms and for confirming their effects on lipid metabolism and hypoxia response. Additionally, exploring the synergistic effects of these bioactive compounds may provide insights into more effective combination therapies for hypoxia-related conditions. Identifying and harnessing the bioactive components within MEs will be instrumental in fully exploiting the therapeutic potential of velvet antler extracts.

## 5. Conclusions

In conclusion, this study elucidates how MEs counteract hypoxia-induced damage in *C. elegans*, particularly through *ech-8*’s role in lipid metabolism reprogramming via the *hif-1* pathway. It not only expands our understanding of velvet antler’s protective effects against hypoxia but also highlights the importance of integrating traditional herbal medicines with modern scientific research. The potential of MEs to reprogram lipid metabolism via *hif-1* presents a novel therapeutic avenue, promising new treatments for hypoxia-related conditions and emphasizing the therapeutic potentials of traditional herbal medicines, especially velvet antler, in protecting against hypoxic stress and related disorders.

## Figures and Tables

**Figure 1 nutrients-16-02257-f001:**
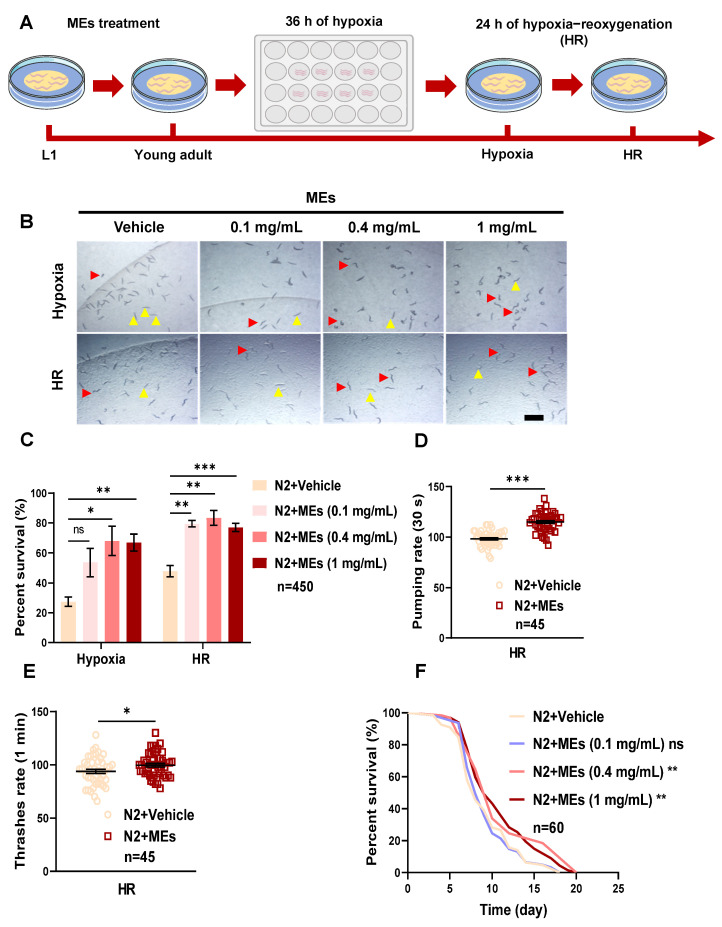
MEs attenuated damage induced by hypoxia in *C. elegans.* (**A**) Illustrated outline of the treatment process involving hypoxia followed by HR. N2 worms were treated with 0, 0.1, 0.4, and 1 mg/mL MEs from the L1 larval stage. (**B**) Images of N2 worms treated with 0, 0.1, 0.4, and 1 mg/mL MEs after hypoxia and HR survival (yellow: deceased worms; red: surviving worms), scale bar represents 2mm. (**C**) Survival rates of worms after hypoxia or HR treatment as shown in (**B**). (**D**) The pharyngeal pumping rate of N2 worms (*n* = 45) treated with or without MEs after HR. (**E**) Body bends of N2 worms (*n* = 45) treated with or without MEs after HR. (**F**) Survival fraction curves of N2 worms (*n* = 60) treated with or without MEs after HR. Error bars represent the SEM of three independent replicates. ns, not significant; *, *p* < 0.05; **, *p* < 0.01; ***, *p* < 0.001. *n* indicates the total number of detected worms in three experimental replicates.

**Figure 2 nutrients-16-02257-f002:**
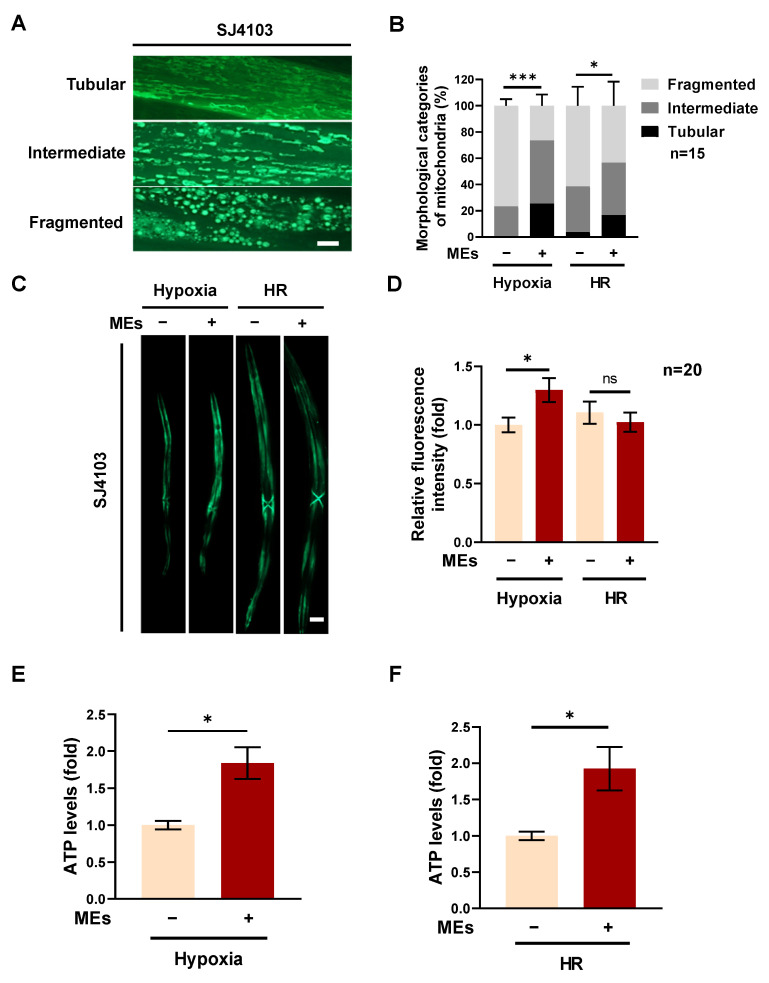
MEs enhanced mitochondrial function in *C. elegans* following hypoxia. (**A**) Representative fluorescent images of MYO-3::GFP fluorescence states in SJ4103 worms. The worms were treated with 0.4 mg/mL MEs from the L1 stage. Worms after hypoxia and HR were collected for fluorescence microscopy imaging; scale bar represents 10 μm. (**B**) Quantitative analysis of mitochondrial morphology as shown in (**A**). The morphological categories of mitochondria were defined as follows: (1) tubular: most mitochondria were interconnected and elongated like a tube shape; (2) intermediate: a combination of interconnected and fragmented mitochondria; (3) fragmented: a majority of round or short mitochondria. (**C**) Representative fluorescence image of SJ4103 worms. Worms were treated with or without 0.4 mg/mL MEs from the L1 stage and worms after hypoxia, and HR were collected for imaging; scale bar indicates 100 μm. (**D**) Quantitative analysis of the intensity of MYO-3::GFP fluorescence shown in (**C**). (**E**,**F**) The ATP content of N2 worms after hypoxia (**E**) and HR (**F**). The worms were treated with or without 0.4 mg/mL MEs from the L1 stage. Error bars represent the SEM of three independent replicates. ns, not significant; *, *p* < 0.05; ***, *p* < 0.001. *n* indicates the total number of detected worms in three experimental replicates.

**Figure 3 nutrients-16-02257-f003:**
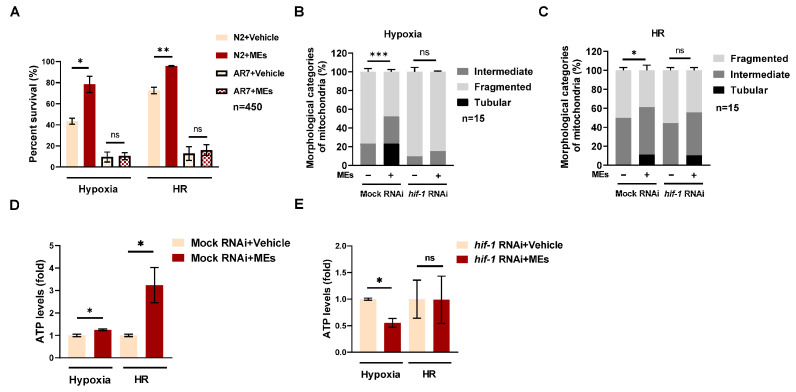
MEs protected against hypoxia damage by HIF-1 in *C. elegans*. (**A**) The survival fraction of N2 and AR7 (*hif-1* mutant) worms (*n* > 450) treated with or without MEs after hypoxia and HR. (**B**,**C**) Quantitative analysis of mitochondrial morphology in SJ403 worms treated with or without MEs following *hif-1* RNAi after hypoxia (**B**) or HR (**C**). Worms at the L1 stage were exposed to either mock RNAi or *hif-1* RNAi, treated with or without MEs, and mitochondrial morphology was analyzed after hypoxia (**B**) or HR (**C**) by fluorescence microscopy. (**D**,**E**) Levels of ATP in N2 worms after hypoxia and HR. The worms were incubated on either mock RNAi (**D**) or *hif-1* RNAi (**E**) NGM plates with or without 0.4 mg/mL MEs from the L1 stage. Error bars represent the SEM of three independent replicates. ns, not significant; *, *p* < 0.05; **, *p* < 0.01; ***, *p* < 0.001. *n* indicates the total number of detected worms in three experimental replicates.

**Figure 4 nutrients-16-02257-f004:**
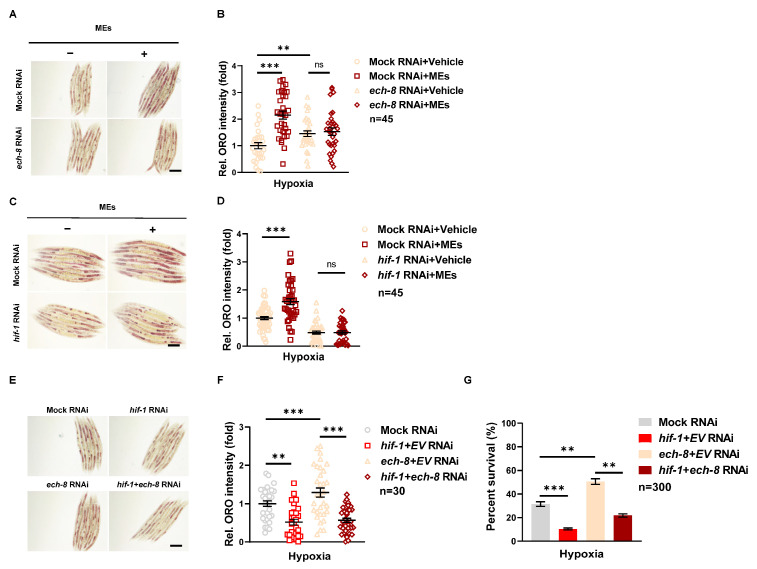
MEs protected against hypoxia damage was dependent on lipid accumulation via *ech-8 and hif-1***.** (**A**) The Oil Red O staining in N2 worms treated with or without MEs following *ech-8* RNAi after hypoxia. The L1 stage worms were exposed to 0.4 mg/mL MEs in mock RNAi NGM plates or *ech-8* RNAi NGM plates, and the Oil Red O staining of worms was checked by microscopy after hypoxia. (**B**) Quantitative analysis of Oil Red O as shown in (**A**). (**C**) The Oil Red O staining in N2 worms treated with or without MEs following *hif-1* RNAi after hypoxia. The L1 stage worms were exposed to 0.4 mg/mL MEs in mock RNAi NGM plates or *hif-1* RNAi NGM plates, and the Oil Red O staining of worms was checked by microscopy after hypoxia. (**D**) Quantitative analysis of Oil Red O as shown in (**C**). (**E**) The Oil Red O staining in N2 worms following *ech-8* RNAi, *hif-1* RNAi or *ech-8*, and *hif-1* double RNAi after hypoxia. The L1 stage worms were exposed to RNAi treatment, and the Oil Red O staining of worms was checked by microscopy after hypoxia. (**F**) Quantitative analysis of Oil Red O as shown in (**E**). (**G**) The survival fraction of N2 worms (*n* > 350) exposed to *ech-8* RNAi, *hif-1* RNAi or *ech-8*, and *hif-1* double RNAi after hypoxia. Error bars represent the SEM of three independent replicates. ns, not significant; **, *p* < 0.01; ***, *p* < 0.001. EV indicates the empty vector of L4440. *n* indicates the total number of detected worms in three experimental replicates.

**Table 1 nutrients-16-02257-t001:** Sequences of qPCR primers.

Gene	Forward Primer Sequence	Reverse Primer Sequence
*cdc-42*	ctgctggacaggaagattacg	ctcggacattctcgaatgaag
*y45f10d.4*	gtcgcttcaaatcagttcagc	gttcttgtcaagtgatccgaca
*hif-1*	ttaacagtcccccgagttgc	gcttccgatgactgggttga
*ech-8*	ttgaacgatcaggatgccgt	ctcatcggaccaccagtagc

## Data Availability

The original contributions presented in the study are included in the article. Further inquiries can be directed to the corresponding author.

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
