# Peer review of "Protective Effects of Velvet Antler Methanol Extracts on Hypoxia-Induced Damage in Caenorhabditis elegans through HIF-1 and ECH-8 Mediated Lipid Accumulation"

_nutrients, 2024, doi:10.3390/nu16142257_

Round 1

Reviewer 1 Report

Comments and Suggestions for Authors

The present article "Protective Effects of Velvet Antler Methanol Extracts on Hypoxia-Induced Damage in Caenorhabditis elegans through HIF-1 and ECH-8 Mediated Lipid Accumulation" elucidates how MEs counteract hypoxia-induced damage in C. elegans, particularly through the role of ech-8 in reprogramming lipid metabolism via the hif-1 pathway. It also highlights the importance of integrating traditional herbal medicines with modern scientific research and demonstrates that methanol extracts from velvet antler (MEs) enhance the survival of C. elegans under hypoxic conditions.

1 – Introduction, 2nd paragraph: Velvet Antler is a type of cartilaginous tissue that covers the growing antlers of deer species such as red deer, elk, and moose, put this information in the paragraph.

2 – Introduction, 3rd paragraph: In addition to the gene ech-8, please include:

·         Gene Function: Description of the gene's biological functions.

·         Gene Structure: Information about the gene's structure, including the number of exons, introns, and relevant structural characteristics.

·         Gene Expression Details: Details on the gene's expression in different developmental stages, physiological, and pathological conditions.

·         Known Gene Variants: Information on known gene variants, including mutations and polymorphisms.

3 – Materials and Methods, 2.1. Strains: C. elegans were grown on Nematode Growth Medium (NGM) with Escherichia coli bacteria. It is known that culturing media like NGM can be contaminated with Mycoplasma due to their small size and lack of a cell wall, potentially affecting biological experiments. It is crucial to detect and eliminate Mycoplasma to ensure research integrity. How was the quality control conducted to ensure there were no contaminants like Mycoplasma in this medium? Was detection done using commercial kits or PCR targeting constitutive Mycoplasma genes?

4 – Materials and Methods, ATP Measurement: The article states, "The resulting supernatant was collected after centrifugation (12,000 rpm, 5 min, 4°C), and ATP concentrations were determined using an enhanced luciferase-based ATP assay kit (Beyotime Biotechnology, S0027, Shanghai, China), following the manufacturer's instructions." Please specify the wavelength used for measurement in a spectrophotometer. Also, provide the wavelength used for protein concentration measurements.

5 – Materials and Methods, 2.7. RNA Isolation and Quantitative PCR (qPCR): Please specify how RNA quality control was performed (including the type of spectrophotometer used), and the RNA quality ratio 260/280. Also, describe the method used for RNA concentration measurement.

6 – Results, Table 1: Sequences of qPCR primers: Please provide the amplicon size in base pairs for each primer pair.

7 – Materials and Methods, 2.8. Statistical analysis lines 120-134: It was reported that continuous variables, including parasitemia, age, weight, and height, were compared between the two sites using a t-test for normally distributed data or the Wilcoxon rank sum test for non-normally distributed data. Include the dilution of the antibodies. Specify which test was used to check the normality of the data. Remember that you can only use parametric tests if your data is normally distributed.

8 – Materials and Methods, 2.9. RNA Interference (RNAi) Assay: In addition to E. coli HT115 with an empty L4440 plasmid used as a negative control in the RNAi experiment, other types of controls such as dsMock (double-stranded Mock), dsControl (double-stranded Control), and dsGFP (double-stranded Green Fluorescent Protein) should be used as nonspecific controls to verify whether observed effects are due to specific gene silencing or nonspecific RNAi effects.

9 – Materials and Methods, Statistical Analysis: The paragraph incorrectly suggests conducting ANOVA analysis without subjecting the data to normality tests (Shapiro-Wilk Test; Kolmogorov-Smirnov Test). Normality testing can only be performed with a minimum of 3 to 5 replicates (test rigor). Please revise this paragraph accordingly.

10 – Results, Figure 1: Include a scale bar on Figure 1B.

11 – Results, Figure 4: Increase the size of Figure 4 as the figures are currently too small to clearly see the details.

12 – Discussion: Enhance the Discussion by relating findings to other studies. Remember that Discussion is not merely a repetition of results. The current discussion lacks...

Reviewer 2 Report

Comments and Suggestions for Authors

Dear Editor and Authors,

The manuscript ‘Protective Effects of Velvet Antler Methanol Extracts on Hypoxia-Induced Damage in Caenorhabditis elegans through HIF-1 and ECH-8 Mediated Lipid Accumulation’ by Ru Li 1#, Hongyuan Li 2#, Xiaohui Wang 2,3* and Yinghua Peng 1 is a research manuscript on velvet antler extract protective effect on C. elegans after low oxygen stress. The Authors concluded that velvet antler extract (MEs) enhance the survival of C. elegans under hypoxic conditions through the stabilization of hypoxia-inducible factor-1 (HIF-1) and the promotion of lipid accumulation, both of which are crucial for mitigating cellular damage.

30 references were collected on the topic. The Figures are of good quality.

M&M

How and when was the methanol extract of velvet antler added to C. elegans- it should be explained

How was the MEs prepared? What was its origin? Was the methanol removed and how?

What was the composition of MEs?

Discussion

More discussion with literature is necessary. In all the discussion only 3 references are cited and only in one sentence about the velvet antler properties and usage in East Asian medicinal practices. Please discuss your results.

Cordially

Round 2

Reviewer 1 Report

Comments and Suggestions for Authors

ok

Reviewer 2 Report

Comments and Suggestions for Authors

Dear Editor and Authors,

The manuscript 'Protective Effects of Velvet Antler Methanol Extracts on Hypoxia-Induced Damage in Caenorhabditis elegans through HIF-1 and ECH-8 Mediated Lipid Accumulation' by  Ru Li , Hongyuan Li 2, Xiaohui Wang and Yinghua Peng is improved after the review,

Cordially